# KG-GPT: A General Framework for Reasoning on Knowledge Graphs Using Large Language Models

**Jiho Kim[1], Yeonsu Kwon[1], Yohan Jo[2], Edward Choi[1]**
[1]KAIST [2]Seoul National University
{jiho.kim, yeonsu.k, edwardchoi}@kaist.ac.kr
yohan.jo@snu.ac.kr

## Abstract

While large language models (LLMs) have made considerable advancements in understanding and generating unstructured text, their application in structured data remains underexplored. Particularly, using LLMs for complex reasoning tasks on knowledge graphs (KGs) remains largely untouched. To address this, we propose KG-GPT, a multi-purpose framework leveraging LLMs for tasks employing KGs. KG-GPT comprises three steps: Sentence Segmentation, Graph Retrieval, and Inference, each aimed at partitioning sentences, retrieving relevant graph components, and deriving logical conclusions, respectively. We evaluate KG-GPT using KG-based fact verification and KGQA benchmarks, with the model showing competitive and robust performance, even outperforming several fully-supervised models. Our work, therefore, marks a significant step in unifying structured and unstructured data processing within the realm of LLMs.[1]

## 1 Introduction

The remarkable advancements in large language models (LLMs) have notably caught the eye of scholars conducting research in the field of natural language processing (NLP) (Brown et al., 2020; Chowdhery et al., 2022; OpenAI, 2023a,b; Anil et al., 2023). In their endeavor to create LLMs that can mirror the reasoning capabilities inherent to humans, past studies have primarily centered their attention on unstructured textual data. This includes, but is not limited to, mathematical word problems (Miao et al., 2020; Cobbe et al., 2021; Patel et al., 2021), CSQA (Talmor et al., 2019), and symbolic manipulation (Wei et al., 2022). While significant strides have been made in this area, the domain of structured data remains largely unexplored.

Structured data, particularly in the form of knowledge graphs (KGs), serves as a reservoir of interconnected factual information and associations, articulated through nodes and edges. The inherent structure of KGs offers a valuable resource that can assist in executing complex reasoning tasks, like multi-hop inference. Even with these advantages, to the best of our knowledge, there is no general framework for performing KG-based tasks (*e.g.* question answering, fact verification) using auto-regressive LLMs.

To this end, we propose a new general framework, called KG-GPT, that uses LLMs' reasoning capabilities to perform KG-based tasks. KG-GPT is similar to StructGPT (Jiang et al., 2023) in that both reason on structured data using LLMs. However, unlike StructGPT which identifies paths from a seed entity to the final answer entity within KGs, KG-GPT retrieves the entire sub-graph and then infers the answer. This means KG-GPT can be used not only for KGQA but also for tasks like KG-based fact verification.

KG-GPT consists of three steps: 1) Sentence (Claim / Question) Segmentation, 2) Graph Retrieval, and 3) Inference. During Sentence Segmentation, a sentence is partitioned into discrete sub-sentences, each aligned with a single triple (*i.e.* [head, relation, tail]). The subsequent step, namely Graph Retrieval, retrieves a potential pool of relations that could bridge the entities identified within the sub-sentences. Then, a candidate pool of evidence graphs (*i.e.* sub-KG) is obtained using the retrieved relations and the entity set. In the final step, the obtained graphs are used to derive a logical conclusion, such as validating a given claim or answering a given question.

To evaluate KG-GPT, we employ KG-based fact verification and KGQA benchmarks, both demanding complex reasoning that utilizes structured knowledge of KGs. In KG-based fact verification, we use FACTKG (Kim et al., 2023), which includes

---

[1]Code is available at https://github.com/jiho283/KG-GPT.

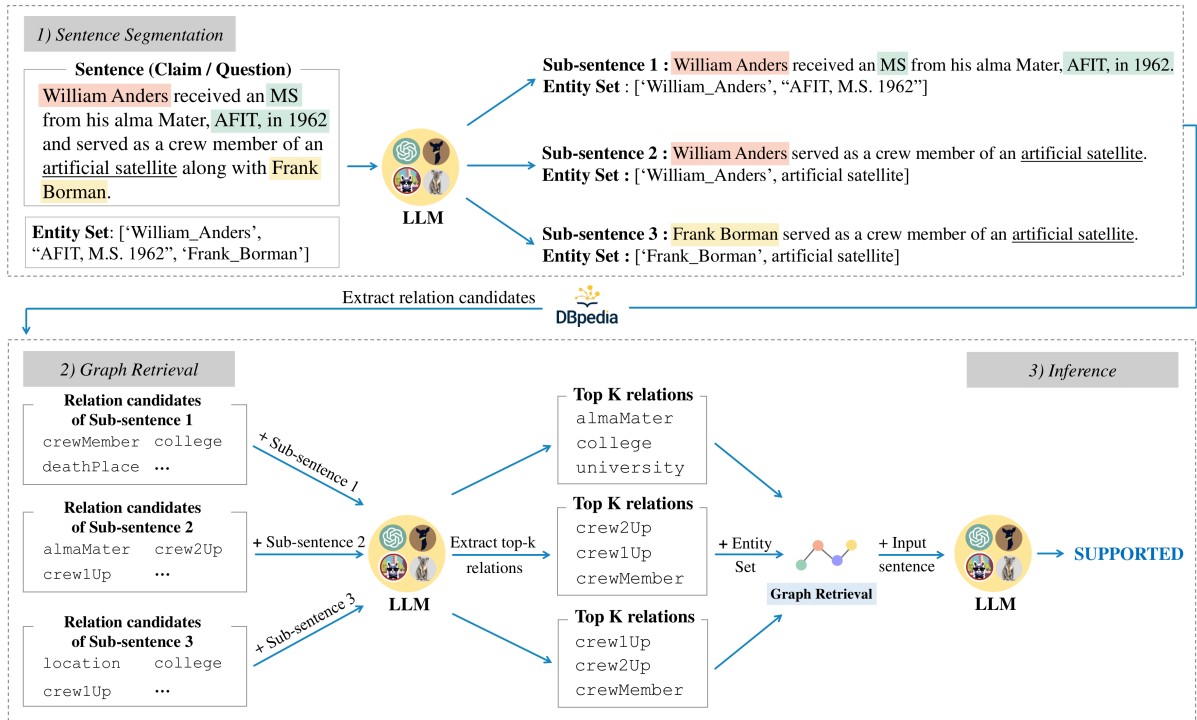

Figure 1: An overview of KG-GPT. The framework comprises three distinct phases: Sentence Segmentation, Graph Retrieval, and Inference. The given example comes from FACTKG. It involves a 2-hop inference from 'William_Anders' to 'Frank_Borman', requiring verification through an evidence graph consisting of three triples. Both 'William_Anders' and 'Frank_Borman' serve as internal nodes in DBpedia (Lehmann et al., 2015), while "AFIT, M.S. 1962" acts as a leaf node. Moreover, *artificial satellite* represents the *Type* information absent from the provided entity set.

various graph reasoning patterns, and KG-GPT shows competitive performance compared to other fully-supervised models, even outperforming some. In KGQA, we use MetaQA (Zhang et al., 2018), a QA dataset composed of 1-hop, 2-hop, and 3-hop inference tasks. KG-GPT shows performance comparable to fully-supervised models. Notably, the performance does not significantly decline with the increase in the number of hops, demonstrating its robustness.

## 2 Method

KG-GPT is composed of three stages: Sentence Segmentation, Graph Retrieval, and Inference, as described in Fig. 1.

We assume a graph $\mathcal{G}$ (knowledge graph consisting of entities $\mathcal{E}$ and relations $\mathcal{R}$), a sentence $S$ (claim or question), and all entities involved in $S$, $\mathcal{E}_S \subset \mathcal{E}$ are given. In order to derive a logical conclusion, we need an accurate evidence graph $\mathcal{G}_E \subset \mathcal{G}$, which we obtain in two stages, Sentence Segmentation and Graph Retrieval. Furthermore, all the aforementioned steps are executed employ-

ing the in-context learning methodology to maximize the LLM's reasoning ability. The prompts used for each stage are in Appendix A.

### 2.1 Sentence Segmentation

Many KG-based tasks require multi-hop reasoning. To address this, we utilize a Divide-and-Conquer approach. By breaking down a sentence into sub-sentences that correspond to a single relation, identifying relations in each sub-sentence becomes easier than finding $n$-hop relations connected to an entity from the original sentence all at once.

We assume $S$ can be broken down into sub-sentences: $S_1, S_2, ..., S_n$ where $S_i$ consists of a set of entities $\mathcal{E}_i \subset \mathcal{E}$ and a relation $r_i \in \mathcal{R}$. Each $e_i^{(j)} \in \mathcal{E}_i$ can be a concrete entity (*e.g. William Anders* in Fig. 1-(1)), or a type (*e.g. artificial satellite* in Fig. 1-(1)). $r_i$ can be mapped to one or more items in $\mathcal{R}$, as there can be multiple relations with similar semantics (*e.g. birthPlace, placeOfBirth*).

### 2.2 Graph Retrieval

To effectively validate a claim or answer a question, it is crucial to obtain an evidence graph (*i.e.* sub-

KG) that facilitates logical conclusions. In this stage, we first aim to retrieve the corresponding relations for each sub-sentence $S_i$ to extract $\mathcal{G}_E$.

For each $S_i$, we use the LLM to map $r_i$ to one or more items in $\mathcal{R}$ as accurately as possible. To do so, we first define $\mathcal{R}_i \subset \mathcal{R}$, which is a set of relations connected to all $e_i^{(j)} \in \mathcal{E}_i$ according to the schema of $\mathcal{G}$ (*i.e.* relation candidates in Fig. 1-(2)). This process considers both the relations connected to a specific entity and the relations associated with the entity's type in $\mathcal{G}$. We further elaborate on the process in Appendix B. Then, we feed $S_i$ and $\mathcal{R}_i$ to the LLM to retrieve the set of final top-K relations $\mathcal{R}_{i,k}$. In detail, relations in $\mathcal{R}_i$ are linearized (*e.g.* [*location, birthYear, ..., birthDate*]) and combined with the corresponding sub-sentence $S_i$ to establish prompts for the LLM and the LLM generates $\mathcal{R}_{i,k} = \{r_i^{(1)}, ..., r_i^{(k)}\}$ as output. In the final graph retrieval step, we can obtain $\mathcal{G}_E$, made up of all triples whose relations come from $\mathcal{R}_{i,k}$ and whose entities come from $\mathcal{E}_i$ across all $S_i$.

## 2.3 Inference

Then, we feed $S$ and $\mathcal{G}_E$ to the LLM to derive a logical conclusion. In order to represent $\mathcal{G}_E$ in the prompt, we linearize the triples associated with $\mathcal{G}_E$ (*i.e.* [[$head_1$, $rel_1$, $tail_1$], ..., [$head_m$, $rel_m$, $tail_m$]]), and then concatenate these linearized triples with the sentence $S$. In fact verification, the determination of whether $S$ is supported or refuted is contingent upon $\mathcal{G}_E$. In question answering, the LLM identifies an entity in $\mathcal{G}_E$ as the most probable answer to $S$.

## 3 Experiments

We evaluate our framework on two tasks that require KG grounding: fact-verification and question-answering. A detailed description of experimental settings can be found in Appendix C.

### 3.1 Dataset

#### 3.1.1 FACTKG

FACTKG (Kim et al., 2023) serves as a practical and challenging dataset meticulously constructed for the purpose of fact verification, employing a knowledge graph for validation purposes. It encompasses 108K claims that can be verified via DBpedia (Lehmann et al., 2015), which is one of the available comprehensive knowledge graphs. These claims are categorized as either *Supported* or *Refuted*. FACTKG embodies five diverse types of

reasoning that represent the intrinsic characteristics of the KG: One-hop, Conjunction, Existence, Multi-hop, and Negation. To further enhance its practical use, FACTKG integrates claims in both colloquial and written styles. Examples of claims from FACTKG can be found in Appendix D.

#### 3.1.2 MetaQA

MetaQA (Zhang et al., 2018) is a carefully curated dataset intended to facilitate the study of question-answering that leverages KG-based approaches in the field of movies. The dataset encompasses over 400K questions, including instances of 1-hop, 2-hop, and 3-hop reasoning. Additionally, it covers a diverse range of question styles. Examples of questions from MetaQA can be found in Appendix D.

### 3.2 Baselines

For evaluation on FACTKG, we use the same baselines as in Kim et al. (2023). These baselines are divided into two distinct categories: *Claim Only* and *With Evidence*. In the *Claim Only* setting, the models are provided only with the claim as their input and predict the label. For this setting, in addition to the existing baselines, we implement a 12-shot ChatGPT (OpenAI, 2023b) baseline. In the *With Evidence* scenario, models consist of an evidence graph retriever and a claim verification model. We employ the KG version of GEAR (Zhou et al., 2019) as a fully supervised model.

In our exploration of MetaQA, we use a selection of prototypical baselines well-known in the field of KGQA. These include models such as KV-Mem (Xu et al., 2019), GraftNet (Sun et al., 2018), EmbedKGQA (Saxena et al., 2020), NSM (He et al., 2021), and UniKGQA (Jiang et al., 2022), which operate in a fully supervised fashion. Additionally, we implement a 12-shot ChatGPT baseline.

## 4 Results & Analysis

### 4.1 FACTKG

We evaluated the models' prediction capability for labels (*i.e.* *Supported* or *Refuted*) and presented the accuracy score in Table 1. As a result, KG-GPT outperforms *Claim Only* models BERT, BlueBERT, and Flan-T5 with performance enhancements of 7.48%, 12.75%, and 9.98% absolute, respectively. It also outperforms 12-shot ChatGPT by 4.20%. These figures emphasize the effectiveness of our framework in extracting the necessary evidence for claim verification, highlighting the positive impact

| Input Type | Training Strategy | Methods | Accuracy |
|---|---|---|---|
| Claim Only | full | BERT | 65.20 |
| | | BlueBERT | 59.93 |
| | zero-shot | Flan-T5 | 62.70 |
| | 12-shot | ChatGPT | 68.48 |
| With Evidence | full | GEAR | **77.65** |
| | 12-shot | | **72.68** |
| | 8-shot | KG-GPT | 67.68 |
| | 4-shot | | 59.53 |

Table 1: The performance of the models on FACTKG. Except for ChatGPT and KG-GPT, all performances are obtained from Kim et al. (2023).

| Training Strategy | Methods | MetaQA 1-hop | MetaQA 2-hop | MetaQA 3-hop |
|---|---|---|---|---|
| full | KV-Mem | 96.2 | 82.7 | 48.9 |
| | GraftNet | 97.0 | 94.8 | 77.7 |
| | EmbedKGQA | **97.5** | 98.8 | 94.8 |
| | NSM | 97.1 | **99.9** | 98.9 |
| | UniKGQA | **97.5** | 99.0 | **99.1** |
| 12-shot | ChatGPT | 60.0 | 23.0 | 38.7 |
| 12-shot | | **96.3** | **94.4** | **94.0** |
| 8-shot | KG-GPT | 95.8 | 93.8 | 68.8 |
| 4-shot | | 94.7 | 92.8 | 46.6 |

Table 2: The performance of the models on MetaQA (Hits@1). The best results for each task and those of 12-shot KG-GPT are in bold.

| Stage | FactKG | MetaQA 1-hop | MetaQA 2-hop | MetaQA 3-hop |
|---|---|---|---|---|
| *Sentence Segmentation* | 39 | 3 | 63 | 100 |
| *Graph Retrieval* | 17 | 4 | 3 | 0 |
| *Inference* | 44 | 93 | 34 | 0 |

Table 3: Number of errors from 100 incorrect samples across each dataset.

Interestingly, the performance of KG-GPT closely matches that of a fully-supervised model. Particularly, it surpasses KV-Mem by margins of 0.1%, 11.7%, and 45.1% across three distinct tasks respectively, signifying its superior performance. While the overall performance of KG-GPT is similar to that of GraftNet, a noteworthy difference is pronounced in the 3-hop task, wherein KG-GPT outperforms GraftNet by 16.3%. The qualitative results including the graphs retrieved by KG-GPT are in Appendix E.2.

### 4.3 Error Analysis

In both FACTKG and MetaQA, there are no corresponding ground truth graphs containing seed entities. This absence makes a quantitative step-by-step analysis challenging. Therefore, we carried out an error analysis, extracting 100 incorrect samples from each dataset: FACTKG, MetaQA-1hop, MetaQA-2hop, and MetaQA-3hop. Table 3 shows the number of errors observed at each step. Notably, errors during the graph retrieval phase are the fewest among the three steps. This suggests that once sentences are correctly segmented, identifying relations within them becomes relatively easy. Furthermore, a comparative analysis between MetaQA-1hop, MetaQA-2hop, and MetaQA-3hop indicates that as the number of hops increases, so does the diversity of the questions. This heightened diversity in turn escalates the errors in Sentence Segmentation.

### 4.4 Ablation Study

#### 4.4.1 Number of In-context Examples

The results for the 12-shot, 8-shot, and 4-shot from the FACTKG and MetaQA datasets are reported in Table 1 and Table 2, respectively. Though there was a predicted improvement in performance with the increase in the number of shots in both FACTKG and MetaQA datasets, this was not uniformly observed across all scenarios. Notably, MetaQA demonstrated superior performance, ex-

of the sentence segmentation and graph retrieval stages. The qualitative results including the graphs retrieved by KG-GPT are in Appendix E.1.

Nonetheless, when compared to GEAR, a fully supervised model built upon KGs, KG-GPT exhibits certain limitations. KG-GPT achieves an accuracy score of 72.68%, which is behind GEAR's 77.65%. This performance gap illustrates the obstacles encountered by KG-GPT in a few-shot scenario, namely the difficulty in amassing a sufficient volume of information from the restricted data available. Hence, despite the notable progress achieved with KG-GPT, there is clear room for improvement to equal or surpass the performance of KG-specific supervised models like GEAR.

### 4.2 MetaQA

The findings on MetaQA are presented in Table 2. The performance of KG-GPT is impressive, scoring 96.3%, 94.4%, and 94.0% on 1-hop, 2-hop, and 3-hop tasks respectively. This demonstrates its strong ability to generalize from a limited number of examples, a critical trait when handling real-world applications with varying degrees of complexity.

ceeding 90%, in both the 1-hop and 2-hop scenarios, even with a minimal set of four examples. In contrast, in both the FACTKG and MetaQA 3-hop scenarios, the performance of the 4-shot learning scenario was similar to that of the baselines which did not utilize graph evidence. This similarity suggests that LLMs may struggle to interpret complex data features when equipped with only four shots. Thus, the findings highlight the importance of formulating in-context examples according to the complexity of the task.

### 4.4.2 Top-K Relation Retrieval

Table 11 shows the performance according to the value of $k$ in FACTKG. As a result, performance did not significantly vary depending on the value of $k$. Table 12 illustrates the average number of triples retrieved for both supported and refuted claims, depending on $k$. Despite the increase in the number of triples as the value of $k$ grows, it does not impact the accuracy. This suggests that the additional triples are not significantly influential.

In MetaQA, the performance and the average number of retrieved triples are also depicted in Table 13 and Table 14, respectively. Unlike the FACTKG experiment, as the value of $k$ rises in MetaQA, it appears that more significant triples are retrieved, leading to improved performance.

## 5 Conclusion

We suggest KG-GPT, a versatile framework that utilizes LLMs for tasks that use KGs. KG-GPT is divided into three stages: Sentence Segmentation, Graph Retrieval, and Inference, each designed for breaking down sentences, sourcing related graph elements, and reaching reasoned outcomes, respectively. We assess KG-GPT's efficacy using KG-based fact verification and KGQA metrics, and the model demonstrates consistent, impressive results, even surpassing a number of fully-supervised models. Consequently, our research signifies a substantial advancement in combining structured and unstructured data management in the LLMs' context.

## Limitations

Our study has two key limitations. Firstly, KG-GPT is highly dependent on in-context learning, and its performance varies significantly with the number of examples provided. The framework struggles particularly with complex tasks when there are in-

sufficient or low-quality examples. Secondly, despite its impressive performance in fact-verification and question-answering tasks, KG-GPT still lags behind fully supervised KG-specific models. The gap in performance highlights the challenges faced by KG-GPT in a few-shot learning scenario due to limited data. Future research should focus on optimizing language models leveraging KGs to overcome these limitations.

## Acknowledgements

This work was supported by Institute of Information & Communications Technology Planning & Evaluation (IITP) grant (No.2019-0- 00075), National Research Foundation of Korea (NRF) grant (NRF-2020H1D3A2A03100945), and the Korea Health Industry Development Institute (KHIDI) grant (No.HR21C0198), funded by the Korea government (MSIT, MOHW).

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

## A Prompts

The prompts for Sentence Segmentation, Graph Retrieval, and Inference can be found in Table 4, Table 5 and Table 6, respectively.

## B Relation Candidates Extraction Algorithm

### B.1 FACTKG

In FACTKG, we develop a new KG called Type-DBpedia. This graph comprises types found in DBpedia and connects them through relations, thereby enhancing the usability of KG content. We describe the detailed process of incorporating $\mathcal{R}_i$ using DBpedia and TypeDBpedia in Algorithm 1.

We denote the entities as $E_1$ and $E_2$ because the sub-sentence always includes two entities in FACTKG. Relations $(e, DBpedia)$ represents the set of relations connected to $e$ in $DBpedia$. Similarly, Relations $(T, TypeDBpedia)$ represents the set of relations connected to $T$ in $TypeDBpedia$.

---

**Algorithm 1:** Extract Relation Candidates

**Input:** Entity Set $E = \{E_1, E_2\}$,
$\quad\quad DBpedia, TypeDBpedia$
**Output:** Relation Candidates $R_i$
Initialization: $T = \emptyset, R_T = \emptyset, R_E = \emptyset$,
$\quad R_i = \emptyset$
**for** *each entity e in E* **do**
  **if** *isType (e)* **then**
    | $T \leftarrow T \cup \{e\}$
  **end**
  **else**
    **if** *isEmpty ($R_E$)* **then**
      | $R_E \leftarrow$ Relations $(e, DBpedia)$
    **end**
    **else**
      | $R_E \leftarrow R_E \cap$ Relations $(e,$
         $DBpedia)$
    **end**
  **end**
**end**
**if** *not isEmpty (T)* **then**
  $R_T \leftarrow$ Relations $(T, TypeDBpedia)$
  $R_i \leftarrow R_E \cap R_T$
**end**
**else**
  | $R_i \leftarrow R_E$
**end**

---

### B.2 MetaQA

For the $n$-hop task in MetaQA, $\mathcal{R}_i$ is constructed from the relations within $n$-hops from the seed entity.

## C Experimental Settings

We utilize ChatGPT[2] (OpenAI, 2023b) across all tasks, and to acquire more consistent responses, we carry out inference with the *temperature* and *top_p* parameters set to 0.2 and 0.1, respectively. For each stage of KG-GPT, 12 pieces of training samples were made into in-context examples and added to the prompt. In FACTKG, there are over 500 existing relations ($|\mathcal{R}| > 500$), so we set $k = 5$ for Top-K relation retrieval. Conversely, in MetaQA, there are only 9 existing relations ($|\mathcal{R}| = 9$), so we set $k = 3$.

## D Data Examples

Examples of data from FACTKG and MetaQA can be found in Tables 7 and 8, respectively.

## E Qualitative Results

### E.1 FACTKG

Table 9 includes the graphs retrieved by KG-GPT, along with the prediction results, for five different claims.

### E.2 MetaQA

Table 10 includes the graphs retrieved by KG-GPT, along with the prediction results, for nine different questions.

## F Top-K Relation Retrieval

### F.1 FACTKG

The performance and the average number of retrieved triples are depicted in Table 11 and Table 12, respectively.

### F.2 MetaQA

The performance and the average number of retrieved triples are depicted in Table 13 and Table 14, respectively.

---

[2]https://platform.openai.com/docs/guides/gpt/chat-completions-api

| Sentence Segmentation Prompt |
|---|
| Please divide the given sentence into several sentences each of which can be represented by one triplet. The generated sentences should be numbered and formatted as follows: #(number). (sentence), (entity set). The entity set for each sentence should contain no more than two entities, with each entity being used only once in all statements. The '##' symbol should be used to indicate an entity set. In the generated sentences, there cannot be more than two entities in the entity set. (i.e., the number of ## must not be larger than two.)

Examples)
Sentence A: Ahmad Kadhim Assad's club is Al-Zawra'a SC.
Entity set: ['Ahmad_Kadhim_Assad' ## "Al-Zawra'a_SC"]
–>Divided:
1. Ahmad Kadhim Assad's club is Al-Zawra'a SC., Entity set: ['Ahmad_Kadhim_Assad' ## "Al-Zawra'a_SC"]

...

Sentence L: An academic journal with code IJPHDE is also Acta Math. Hungar.
Entity set: ["Acta Math. Hungar." ## "IJPHDE"]
–>Divided:
1. An academic journal is with code IJPHDE., Entity set: ['academic journal' ## "IJPHDE"]
2. An academic journal is also Acta Math. Hungar., Entity set: ['academic journal' ## "Acta Math. Hungar."]


Your Task)
Sentence: <<<<CLAIM>>>>
Entity set: <<<<ENTITY_SET>>>>
–>Divided: |

Table 4: Sentence Segmentation Prompt.

| Relation Retrieval Prompt |
|---|
| I will give you a set of words.
Find the top <<<>>>elements from Words set which are most semantically related to the given sentence. You may select up to <<<>>>words. If there is nothing that looks semantically related, pick out any <<<>>>elements and give them to me.

Examples)
Sentence A: Ahmad Kadhim Assad's club is Al-Zawra'a SC.
Words set: ['club', 'clubs', 'parent', 'spouse', 'birthPlace', 'deathYear', 'leaderName', 'awards', 'award', 'vicepresident', 'vicePresident']
Top 2 Answer: ['club', 'clubs']

...

Sentence L: An academic journal with code IJPHDE is also Acta Math. Hungar.
Words set: ['abbreviation', 'placeOfBirth', 'owner', 'coden', 'almaMater', 'dean', 'coach', 'writer', 'firstAired', 'director', 'formerTeam', 'starring', 'birthPlace']
Top 2 Answer: ['abbreviation', 'coden']


Now let's find the top <<<>>>elements.
Sentence: <<<<SENTENCE>>>>
Words set: <<<<RELATION_SET>>>>
Top <<<>>>Answer: |

Table 5: Relation Retrieval Prompt. The prompt is used when retrieving a relation to retrieve a graph.

| Inference Prompt |
|---|
| You should verify the claim based on the evidence set. |
| Each evidence is in the form of [head, relation, tail] and it means "head's relation is tail.". |
| |
| Verify the claim based on the evidence set. (True means that everything contained in the claim is supported by the evidence.) |
| |
| Please note that the unit is not important. (e.g. "98400" is also same as 98.4kg) |
| Choose one of {True, False}, and give me the one-sentence evidence. |
| |
| Examples) |
| |
| Claim A: Ahmad Kadhim Assad's club is Al-Zawra'a SC. |
| Evidence set: [['Ahamad_Kadhim', 'clubs', "Al-Zawra'a SC"]] |
| Answer: True, based on the evidence set, Ahmad Kadhim Assad's club is Al-Zawra'a SC. |
| |
| ... |
| |
| Claim L: The place, designed by Huseyin Butuner and Hilmi Guner, is located in a country, where the leader is Paul Nurse. |
| Evidence set: [["Baku_Turkish_Martyrs'_Memorial", 'designer', "Hüseyin Bütüner and Hilmi Güner"], ["Baku_Turkish_Martyrs'_Memorial", 'location', 'Azerbaijan']] |
| Answer: False, there is no evidence for Paul Nurse. |
| |
| |
| Now let's verify the Claim based on the Evidence set. |
| Claim: <<<<CLAIM>>>> |
| Evidence set: <<<<EVIDENCE_SET>>>> |
| Answer: |

Table 6: Inference Prompt.

| Reasoning Type | Claim Example | Graph |
|---|---|---|
| **One-hop** | AIDAstella was built by Meyer Werft. | $s \xrightarrow{r_2} m$ |
| **Conjunction** | AIDA Cruise line operated the AIDAstella which was built by Meyer Werft. | $c \xleftarrow{r_3} s \xrightarrow{r_2} m$ |
| **Existence** | Meyer Werft had a parent company. | $m \dashrightarrow^{r_1}$ |
| **Multi-hop** | AIDAstella was built by a company in Papenburg. | $s \xrightarrow{r_2} x \xrightarrow{r_4} p$ |
| **Negation** | AIDAstella was not built by Meyer Werft in Papenburg. | $s \xrightarrow{r_2} m \xrightarrow{r_4} p$ |

Table 7: Five different reasoning types of FACTKG. $r_1$: parentCompany, $r_2$: shipBuilder, $r_3$: shipOperator, $r_4$: location, $m$: Meyer Werft, $s$: AIDAstella, $c$: AIDA Cruises.

| Task | Question Examples |
|---|---|
| 1-hop | 1. what does [Helen Mack] star in? |
| | 2. what is the main language in [Karate-Robo Zaborgar]? |
| | 3. who is the writer of [Boyz n the Hood]? |
| 2-hop | 1. who are movie co-directors of [Delbert Mann]? |
| | 2. what genres do the films starred by [Al St. John] fall under? |
| | 3. which films share the screenwriter with [King Arthur]? |
| 3-hop | 1. the films that share directors with the film [Catch Me If You Can] were in which languages? |
| | 2. who are the directors of the movies written by the writer of [She]? |
| | 3. when did the movies release whose actors also appear in the movie [Operator 13]? |

Table 8: Question examples from MetaQA.

| Type | Claim | Retrieved Graph | Prediction |
|---|---|---|---|
| Conjunction | Yes, Agra Airport is located in India where the leader is Narendra Modi. | ['Agra_Airport', location, 'India'], ['India', leader, 'Narendra_Modi'], ['India', leaderName, 'Narendra_Modi'], ['Narendra_Modi', birthPlace, 'India'] | Supported |
| Conjunction | I wasn't aware that 103 Colmore Row, located in Birmingham, with 23 floors, was completed in 1976. | ['103_Colmore_Row', location, 'Birmingham'], ['103_Colmore_Row', floorCount, "23"], ['103_Colmore_Row', completionDate, "1976"], ['103_Colmore_Row', buildingEndDate, "1976"] | Supported |
| Multi-hop | Alfredo Zitarrosa died in a city, Uruguay (which has Raul Fernando Sendic Rodriguez as leader). | ['Alfredo_Zitarrosa', deathPlace, 'Uruguay'], ['Alfredo_Zitarrosa', birthPlace, 'Uruguay'], ['Montevideo', country, 'Uruguay'], ['Alfredo_Zitarrosa', deathPlace, 'Montevideo'], ['Alfredo_Zitarrosa', birthPlace, 'Montevideo'], ['Uruguay', capital, 'Montevideo'], ['Uruguay', leader, 'Raúl_Fernando_Sendic_Rodríguez'], ['Uruguay', leaderName, 'Raúl_Fernando_Sendic_Rodríguez'] | Supported |
| Negation | Al-Taqaddum Air Base is located in Fallujah which is not in Iraq. | ['Al-Taqaddum_Air_Base', city, 'Fallujah'], ['Al-Taqaddum_Air_Base', cityServed, 'Fallujah'], ['Fallujah', country, 'Iraq'] | Refuted |
| Multi-hop | A country is the location of the Adare Manor, is run by leader Enda Kenny and the natives are Irish people. | ['Adare_Manor', country, 'Republic_of_Ireland'], ['Republic_of_Ireland', leader, 'Enda_Kenny'], ['Republic_of_Ireland', leaderName, 'Enda_Kenny'], ['Adare_Manor', locationCountry, 'Republic_of_Ireland'], ['Republic_of_Ireland', demonym, 'Irish_people'] | Supported |

Table 9: Qualitative results from FACTKG.

| Task | Question | Retrieved Graph | Prediction |
|---|---|---|---|
| 1-hop | what films does [Brigitte Nielsen] appear in? | ['Cobra', starred_actors, 'Brigitte Nielsen'], ['Red Sonja', starred_actors, 'Brigitte Nielsen'] | 'Cobra' |
| | can you name a film directed by [Nikolai Müllerschön]? | ['The Red Baron', directed_by, 'Nikolai Müllerschön'] | 'The Red Baron' |
| | what type of film is [Six Shooter]? | ['Six Shooter', has_genre, 'Short'] | 'Short' |
| 2-hop | when did the films starred by [Deborah Van Valkenburgh] release? | ['Mean Guns', starred_actors, 'Deborah Van Valkenburgh'], ['Mean Guns', release_year, '1997'] | '1997' |
| | which films have the same director of [The Duellists]? | ['The Duellists', directed_by, 'Ridley Scott'], ['The Counselor', directed_by, 'Ridley Scott'] | 'The Counselor' |
| | what genres are the movies written by [Robert Kenner] in? | ['Food, Inc.', written_by, 'Robert Kenner'], ['Food, Inc.', has_genre, 'Documentary'] | 'Documentary' |
| 3-hop | what are the genres of the movies whose writers also wrote [The Lives of a Bengal Lancer]? | ['The Lives of a Bengal Lancer', written_by, 'John L. Balderston'], ['Frankenstein', written_by, 'John L. Balderston'], ['Frankenstein', has_genre, 'Horror'] | 'Horror' |
| | when did the movies starred by [Seeking Justice] actors release? | ['Seeking Justice', starred_actors, 'Nicolas Cage'], ['World Trade Center', starred_actors, 'Nicolas Cage'], ['World Trade Center', release_year, '2006'] | '2006' |
| | what types are the movies starred by actors in [A Thin Line Between Love and Hate]? | ['A Thin Line Between Love and Hate', directed_by, 'Martin Lawrence'], ["Big Momma's House", starred_actors, 'Martin Lawrence'], ["Big Momma's House", has_genre, 'Comedy'], ['A Thin Line Between Love and Hate', written_by, 'Martin Lawrence'], ['A Thin Line Between Love and Hate', starred_actors, 'Martin Lawrence'] | 'Comedy' |

Table 10: Qualitative results from MetaQA.

| Input Type | Method | $k$ | Accuracy |
|---|---|---|---|
| *With Evidence* | KG-GPT | 3 | 72.12 |
| | | 5 | **72.68** |
| | | 10 | 72.40 |

Table 11: Performance changes according to the change in the $k$ value in FACTKG.

| $k$ | FactKG | |
|---|---|---|
| | Supported | Refuted |
| 3 | 1.93 | 1.08 |
| 5 | 2.52 | 1.39 |
| 10 | 3.13 | 1.86 |

Table 12: The average number of retrieved triples changes according to the change in the $k$ value in FACTKG.

| Method | $k$ | MetaQA 1hop | MetaQA 2hop | MetaQA 3hop |
|---|---|---|---|---|
| KG-GPT | 1 | 97.0 | 93.4 | 89.4 |
| | 3 | 96.3 | 94.4 | 94.0 |
| | 5 | 96.7 | 95.0 | 93.7 |

Table 13: Performance changes according to the change in the $k$ value in MetaQA.

| $k$ | MetaQA | | |
|---|---|---|---|
| | 1-hop | 2-hop | 3-hop |
| 1 | 2.06 | 4.43 | 3.25 |
| 3 | 2.16 | 4.53 | 3.59 |
| 5 | 2.12 | 4.51 | 3.65 |

Table 14: The average number of retrieved triples changes according to the change in the $k$ value in MetaQA.

| Motivation | | | |
|---|---|---|---|
| *Practical* | *Cognitive* | *Intrinsic* | *Fairness* |
| 4.1 4.2 | | | |

| Generalisation type | | | | | |
|---|---|---|---|---|---|
| *Compositional* | *Structural* | *Cross Task* | *Cross Language* | *Cross Domain* | *Robustness* |
| 4.1 4.2 | | | | | 4.1 4.2 |

| Shift type | | | |
|---|---|---|---|
| *Covariate* | *Label* | *Full* | *Assumed* |
| | | | 4.1 4.2 |

| Shift source | | | |
|---|---|---|---|
| *Naturally occuring* | *Partitioned natural* | *Generated shift* | *Fully generated* |
| 4.1 4.2 | | | |

| Shift locus | | | |
|---|---|---|---|
| *Train–test* | *Finetune train–test* | *Pretrain–train* | *Pretrain–test* |
| 4.1 4.2 | | | |

Table 15: GenBench Evaluation Card from Hupkes et al. (2022).