# OpenReview forum: "KG-GPT: A General Framework for Reasoning on Knowledge Graphs Using Large Language Models"
_EMNLP/2023/Conference — EMNLP 2023 Findings_

### Official Review · Reviewer_Z28W · 2023-08-03

**Soundness:** 3

**Excitement:**

3: Ambivalent: It has merits (e.g., it reports state-of-the-art results, the idea is nice), but there are key weaknesses (e.g., it describes incremental work), and it can significantly benefit from another round of revision. However, I won't object to accepting it if my co-reviewers champion it.

**Paper Topic And Main Contributions:**

This paper proposes KG-GPT, a multi-purpose framework leveraging LLMs for tasks employing KGs. KG-GPT comprises three steps: sentence segmentation, graph retrieval, and inference. The sentence segmentation step partitions a sentence into discrete sub-sentences, each aligned with a single triple. The graph retrieval step retrieves a potential pool of relations within the sub-sentences. In the final step, inference obtains graphs used to derive a logical conclusion, such as validating a given claim or answering a given question.

**Questions For The Authors:**

Q1. My concern regarding this paper pertains to the graph retrieval step in addressing KBQA. In this step, the relation candidates extraction algorithm is employed to extract candidate subgraph relations, which are subsequently filtered through the LLM method to obtain the top-k relations that form the subgraph. However, in this process, how do the authors ensure that the extracted subgraphs contain the answer nodes, especially in the case of multi-hop questions?

Q2. The experiments of this paper only present the performance of ChatGPT in the zero-shot setting. However, my main concern lies in the fact that the proposed KG-GPT does not truly comprehend the graph structure. It appears that for fact verification, KG-GPT relies on its pre-existing knowledge to determine the plausibility of given statements. As for KBQA, KG-GPT first generates answers independently and then selects similar entities from the provided candidate answers. No ablation experiments are provided in the study concerning this issue, nor is there any evidence to suggest that KG-GPT's judgments are based on the provided knowledge graph. The authors should include ChatGPT's experiment results in the same experimental setting but without the extraction of sub-KG.

**Reasons To Accept:**

1. This paper proposes a versatile framework that utilizes LLMs for tasks that use KGs.

2. This paper evaluates KG-GPT using KG-based fact verification and KGQA benchmarks, which fills the gap in complex reasoning on knowledge graphs for LLMs.

**Reasons To Reject:**

1. This paper fails to provide a clear introduction to the sentence segmentation step, which constitutes a crucial aspect.

2. This paper presents a feasible method for combining LLMs with KGs. However, it claims that KG-GPT can comprehend graph structures, yet the experiments provided in the paper fail to substantiate this claim. Additional comparative experiments and case studies are required to support this assertion adequately.

**Reproducibility:**

3: Could reproduce the results with some difficulty. The settings of parameters are underspecified or subjectively determined; the training/evaluation data are not widely available.

**Reviewer Confidence:**

4: Quite sure. I tried to check the important points carefully. It's unlikely, though conceivable, that I missed something that should affect my ratings.

**Typos Grammar Style And Presentation Improvements:**

In the 97th line of the second page of the paper, the author's discussion on sentence segmentation is overly rudimentary. The author should provide a succinct and refined elucidation of the module's workflow at this juncture.

---

> ### Author Rebuttal · Authors · 2023-08-29
>
> Thank you for the time and effort spent in carefully reviewing our work. Please kindly find the responses below.
>
> **Reasons To Reject 1**
>
> Many KG-based tasks require multi-hop reasoning. To address this, problems are tackled using a Divide-and-Conquer approach. By breaking down a sentence into sub-sentences that correspond to a single relation, identifying relations in each sub-sentence becomes easier than finding n-hop relations connected to an entity from the original sentence all at once. Additional explanations will be added to the revised manuscript to clarify.
>
> **Reasons To Reject 2 and Question 2**
>
> | | FactKG | MetaQA-1hop | MetaQA-2hop | MetaQA-3hop |
> |---------|:---------:|:---------:|:---------:|:---------:|
> | 12-shot ChatGPT| 68.48| 60.0| 23.0| 38.7|
> | KG-GPT | 72.68| 96.3| 94.4| 94.0|
>
> We conducted an additional 12-shot ChatGPT experiment. The results are in the table above, where 12-shot ChatGPT demonstrated lower performance than KG-GPT in all cases, proving the positive impact of the retrieved graph. However, as you can see, the performance gap between the two models in FactKG is quite narrow (both perform quite well), whereas in MetaQA, there's a significant difference (KG-GPT is near perfect, while 12-shot ChatGPT is not). There are two reasons for this.
>
> - First, the number of unique relations used in MetaQA’s KG (i.e. WikiMovies) is 9, whereas FactKG’s (i.e. DBPedia) has over 500. Thus, KG-GPT can utilize graphical knowledge much more effectively in MetaQA, where relation retrieval is comparatively easier.
>
> - Second, while MetaQA is a question-answering task, FactKG is a binary classification task. Thus, as you pointed out, it is possible for ChatGPT to make a correct guess based on its pre-existing knowledge. However, in MetaQA, it's crucial to find the correct entity. Without proper multi-step inference based on the KG, it becomes extremely challenging to find the right answer, leading to ChatGPT's lower performance in MetaQA.
>
> - In conclusion, as you've mentioned, in certain situations, it seems possible for LLMs to rely on their prior knowledge to get the right answer. But fundamentally, we've experimentally confirmed that KG-GPT certainly utilizes graphical knowledge. In the future, we will further develop our algorithm to retrieve more precise subgraphs in complex KGs.
>
>
>
> **Question 1**
>
> In graph retrieval, we sequentially visit neighbors from the seed entity using the top $k$ relations to construct a subgraph. If sentence segmentation and relation retrieval for each sub-sentence are done correctly, the answer will inevitably be in the subgraph.
>
>  We will demonstrate the process of extracting a subgraph of a 2-hop question using an example. The subgraph $G$ starts as an empty set, and triples are added according to the algorithm below to obtain the final graph containing the answer node of the question.
>
>  First, we segment the question $S$ (with seed entity $e$) into sub-sentences $S_1$ (containing $e$) and $S_2$. Then, for each $S_1$ and $S_2$, we retrieve the top $k$ relations $R_1 = ${$r_{1}^{1}, r_{1}^{2}, ..., r_{1}^{k}$} and $R_2 = ${$r_{2}^{1}, r_{2}^{2}, ..., r_{2}^{k}$}. For each $r_{1}^{m} \in R_1$, we determine the set of entities connected to $e$ via $r_{1}^{m}$ as $E_{(1, m)} = ${$e_{(1, m)}^{1}, e_{(1, m)}^{2}, ..., e_{(1, m)}^{n}$}. Then, for each $e_{(1, m)}^{p} \in E_{(1, m)}$, we find the entities connected by every $r_{2}^{l} \in R_2$ as $E_{(2, m, l)} = ${$e_{(2, m, l)}^{1}, e_{(2, m, l)}^{2}, ..., e_{(2, m, l)}^{t}$}. If $E_{(2, m, l)}$ is an empty set, subgraph $G$ remains unchanged. If not, we randomly select $e_{(2, m, l)}^{w}$ from $E_{(2, m, l)}$ and add the two triples $[e, r_{1}^{m}, e_{(1, m)}^{p}]$ and $[e_{(1, m)}^{p}, r_{2}^{l}, e_{(2, m, l)}^{w}]$ to $G$. This algorithm is used for every $r_{1}^{m} \in R_1$ and $r_{2}^{l} \in R_2$ pair to complete $G$.
>
>  The mathematical notation might not be intuitively clear, so in the revised manuscript, we will include a figure detailing the steps to help explain.

---

### Official Review · Reviewer_PVWa · 2023-08-04

**Soundness:** 3

**Excitement:**

3: Ambivalent: It has merits (e.g., it reports state-of-the-art results, the idea is nice), but there are key weaknesses (e.g., it describes incremental work), and it can significantly benefit from another round of revision. However, I won't object to accepting it if my co-reviewers champion it.

**Missing References:**

[1] Yasunaga, Michihiro, et al. "QA-GNN: Reasoning with Language Models and Knowledge Graphs for Question Answering." Proceedings of the 2021 Conference of the North American Chapter of the Association for Computational Linguistics: Human Language Technologies. 2021.

[2] Wang, Liang, et al. "SimKGC: Simple contrastive knowledge graph completion with pre-trained language models." arXiv preprint arXiv:2203.02167 (2022).

[3] Agarwal, Oshin, et al. "Knowledge Graph Based Synthetic Corpus Generation for Knowledge-Enhanced Language Model Pre-training." Proceedings of the 2021 Conference of the North American Chapter of the Association for Computational Linguistics: Human Language Technologies. 2021.

[4 Thorne, James, et al. "FEVER: a Large-scale Dataset for Fact Extraction and VERification." Proceedings of the 2018 Conference of the North American Chapter of the Association for Computational Linguistics: Human Language Technologies, Volume 1 (Long Papers). 2018.

**Paper Topic And Main Contributions:**

This paper studies the problem of knowledge graph reasoning using large language models (LLMs), specifically, ChatGPT. A framework, called KG-GPT is proposed to solve this problem. KG-GPT consists of three parts: Sentence Segmentation, Graph Retrieval, and Inference. Sentence Segmentation split a claim or question into several sub-sentences which contain only one triple, Graph Retrieval retrieves the knowledge graph and finds evidence for the claim or question, and Inference do the final decision. All three components are leveraging ChatGPT by providing prompts. The method is evaluated on two datasets: FactKG (fact verification) and MetaQA (question-answering). The experiment results show competitive performance with fully supervised models.

**Questions For The Authors:**

Question A: Why is it necessary to use Large Language Models (LLMs) for sentence segmentation and graph retrieval when there are already methods for relation extraction, entity linking, and graph query and matching?

Question B: What is the rationale behind segmenting a sentence into sub-sentences, each containing only one relation?

Question C: see W2.

**Reasons To Accept:**

S1 The paper designs a framework for reasoning based on knowledge graph. And the framework shows competitive performance, which gives us some insights of using ChatGPT for reasoning based on knowledge graph.
S2 The details of the way of training the model are provided, for instance, the prompts used in every step. This level of transparency enhances the reproducibility and understanding of the proposed approach.

**Reasons To Reject:**

W1 The paper makes several claims that some have found to be factual but not entirely accurate. To strengthen its credibility, a more thorough literature review may be necessary.
---- The paper claims “there is no general framework for performing KG-based tasks (e.g. question answering, fact verification) using LLMs.” However, I found [1] proposes a framework using RoBERTa-large and KG for question-answering. There are also some other tasks like knowledge graph completion and graph-to-text generation which are KG-based tasks use LLMs, for example, [2] and [3], respectively.

-------------------------------------------
Put W1 to presentation improvement.

W2 The formal notations and definitions have various problems related to soundness, clarity, and rigor.
--- line 101-102, “Si consists of a set of entities and a relation”, it is unclear why a sequence of tokens consists of a set of entities and a relation. And in lines 115-118, it mentions Ri is a set of relations, isn’t each sub-sentence only have one relation according to lines 101-102?
--- Algorithm 1, what is TypeDBpedia and how it is gotten; it is unclear what E1 and E2 are. What is “Relations (T, TypeDBpedia)” and “Relations (e, DBpedia)”?

W3 One aspect of the paper's precondition raises concerns for me. It assumes that "all entities involved in S are given," which may not align with real-world applications. In practice, users may not specify all entities in their questions or claims, making this assumption less realistic.

W4 The choice of datasets for the experiments seems atypical. For instance, FactKG, released in 2023, might not have gained widespread adoption yet, whereas [4] FEVER serves as a widely used benchmark for claim verification.

W5 The paper appears to lack technical depth, as all three components primarily rely on providing prompts to ChatGPT.

**Reproducibility:**

4: Could mostly reproduce the results, but there may be some variation because of sample variance or minor variations in their interpretation of the protocol or method.

**Reviewer Confidence:**

4: Quite sure. I tried to check the important points carefully. It's unlikely, though conceivable, that I missed something that should affect my ratings.

**Typos Grammar Style And Presentation Improvements:**

P1 See W1, W2.

P2: I will suggest conducting experiments using all available training instances for ChatGPT to determine if the performance remains consistently good.

P3: Regarding the terms "12-shot," "8-shot," and "4-shot," do they refer to using 12, 8, and 4 instances, respectively, for training the model, or do they represent using 12%, 8%, and 4% of training set instances for training the model?

---

> ### Author Rebuttal · Authors · 2023-08-29
>
> Thank you for the time and effort spent in carefully reviewing our work. Please kindly find the responses below.
>
> **Reasons To Reject W1**
>
> We think the statement “RoBERTa is one of LLMs” is not entirely agreeable, as the term “LLM” typically refers to generative language models (e.g. ChatGPT, GPT-4, LLaMA). Still, we will use “Generative Large Language Model” instead of “Large Language Model” in the revised manuscript. Additionally, it seems unreasonable to put this in the “reasons to reject” section, when there is a more appropriate section “Typos Grammar Style And Presentation Improvements”.
>
>
> **Reasons To Reject W2 and Question C**
>
> $S_i$ is a sequence of tokens that semantically contains the meanings of entities and relations. As shown in lines 104-107, $r_i$ semantically carries a single meaning but can map to more than one relation (e.g. birthPlace, placeOfBirth), thus defining the set $R_i$. $TypeDBpedia$, as depicted in lines 422-425, is a Knowledge Graph connecting types found in $DBpedia$. For instance, the triple [United_States, president, Obama] in $DBpedia$ is transformed into [country, president, person] to compose $TypeDBpedia$. $E_1$ and $E_2$ refer to the entities that appear in the sentence. In FactKG, entities included in the sub-sentence are always two, which is why they are denoted as $E_1$ and $E_2$. $Relations (T, TypeDBpedia)$ represent the set of relations connected to $T$ in $TypeDBpedia$, and $Relations (e, DBpedia)$ represent the set of relations connected to $e$ in $DBpedia$. We will add the explanations in the final version.
>  Additionally, it seems unreasonable to put this in the “reasons to reject” section, when there is a more appropriate section “Typos Grammar Style And Presentation Improvements”.
>
> **Reasons To Reject W3**
>
> In FactKG [1], all the baselines were evaluated in an entity-given setting, and in KGQA, the assumption where a seed entity is provided is commonly used in most KGQA research [2, 3, 4, 5]. Although entity identification/linking is an essential step in any knowledge-based QA, it is rather easily achievable by interacting with the user whenever necessary (e.g. the machine can ask which entity the user was referring to, or let the user choose the intended entity from the entity list), which is far safer than the model trying to guess which entity the user was referring to, simply based on the very short input context. This is why most past KGQA studies and FactKG focus on the logic part, rather than trying to perfect the entity identification/linking part. We will add this discussion in the revised manuscript.
>
>
> **Reasons To Reject W4**
>
> The fact that FactKG is not widespread does not change the core value of our work. We chose to use FactKG because it is the only KG-based fact verification dataset where the claim is in natural language. And it encompasses the most diverse reasoning types among all KG-based fact verification datasets (other datasets typically only require 1-hop inference), so the fact that we tested KG-GPT on FactKG instead of older (but easier) ones should be viewed as a strength of our work, not weakness.
>
>  FEVER uses unstructured text as evidence, so it's not even compatible with the scope of our work, which is giving LLMs the ability to reason based on knowledge graphs.
>
>
>
> **Reasons To Reject W5**
>
> The most significant technical contribution of our research is that it provides the first framework that enables the use of LLMs for KG-based tasks. Within this framework, there are three steps, each bearing technical significance.
>
> - First, in the sentence segmentation step, sentences are divided into simpler sub-sentences that correspond to a single relation. This can be seen as dividing a complex problem into simpler ones using a Divide-and-Conquer strategy.
>
> - Second, in the graph retrieval step, LLM is employed to retrieve relations. Recent studies [6] have used generative language models as document retrievers, and our approach similarly leverages generative models for graph retrieval.
>
> - Third, in the inference step, triples from the provided subgraph are combined to reason and derive the final answer. This process, which involves chaining multiple triples for reasoning, can be viewed as the Knowledge Graph version of Chain-Of-Thought [7].
>
> In this manner, each inherent step holds technical significance beyond mere prompting.
>
>
> **Question A**
>
> It is not our research objective to solve KG-based tasks with whatever means necessary. It is to design a framework to grant the LLM the ability to reason with KG (as we explicitly described in the Abstract and Introduction), which we demonstrated successfully using two complex KG-based datasets.
>
> Moreover, even if you were interested in solving KG-based tasks with whatever means necessary, LLMs still have a clear advantage compared to combining multiple modules each developed for a specific sub-task (e.g. relation extraction, NER, etc.), since the latter typically requires a separate training data and training process for all individual modules. However, using LLMs does not require any of those extra overhead, as you can harness the power of In-Context Learning.
>
> **Question B**
>
> Many KG-based tasks require multi-hop reasoning. To address this, problems are tackled using a Divide and Conquer approach. By breaking down a sentence into sub-sentences that correspond to a single relation, identifying relations in each sub-sentence becomes easier than finding n-hop relations connected to an entity from the original sentence all at once. Additional explanations will be added to the revised manuscript to clarify.
>
> **Typos Grammar Style And Presentation Improvements P3**
>
> Our framework employs In-Context Learning without fine-tuning process and 12-shot, 8-shot, 4-shot refer to 12, 8, and 4 instances, respectively.
>
>
> [1] Kim, Jiho, et al. FactKG: Fact Verification via Reasoning on Knowledge Graphs. In Proceedings of the 61st Annual Meeting of the Association for Computational Linguistics. 2023.
>
> [2] Jin, W., et al. Improving embedded knowledge graph multi-hop question answering by introducing relational chain reasoning. Data Min Knowl Disc 37, 255–288 (2023).
>
> [3] Saxena, Apoorv, et al. Improving Multi-hop Question Answering over Knowledge Graphs using Knowledge Base Embeddings. In Proceedings of the 58th Annual Meeting of the Association for Computational Linguistics. 2020.
>
> [4] He, Gaole, et al. Improving Multi-hop Knowledge Base Question Answering by Learning Intermediate Supervision Signals. In WSDM'2021.
>
> [5] Jiang, Jinhao, et al. "UniKGQA: Unified Retrieval and Reasoning for Solving Multi-hop Question Answering Over Knowledge Graph." The Eleventh International Conference on Learning Representations. 2022.
>
> [6] Zhu, Yutao, et al. Large Language Models for Information Retrieval: A Survey. arXiv preprint arXiv:2308.07107 (2023).
>
> [7] Wei, Jason, et al. "Chain-of-thought prompting elicits reasoning in large language models." Advances in Neural Information Processing Systems 35 (2022): 24824-24837.

---

### Official Review · Reviewer_USSs · 2023-08-04

**Soundness:** 4

**Excitement:**

3: Ambivalent: It has merits (e.g., it reports state-of-the-art results, the idea is nice), but there are key weaknesses (e.g., it describes incremental work), and it can significantly benefit from another round of revision. However, I won't object to accepting it if my co-reviewers champion it.

**Missing References:**

[1] Jiang, J., Zhou, K., Dong, Z., Ye, K., Zhao, W.X. and Wen, J.R., 2023. Structgpt: A general framework for large language model to reason over structured data. arXiv preprint arXiv:2305.09645.

**Paper Topic And Main Contributions:**

This manuscript delves into the problem of knowledge graph reasoning using Large Language Models (LLMs) and proposes a general framework consisting of three steps: sentence segmentation, graph retrieval, and inference. The triples associated with the sub-KG generated in the first two steps are linearized and concatenated with the input sentence. The framework is evaluated on fact-verification and KGQA tasks, and the results demonstrate its effectiveness.

**Questions For The Authors:**

It is worth noting that StructGPT [1] appears to be very similar to KG-GPT, as both studies focus on the KGQA problem and use the same MetaQA data. Therefore, it would be interesting to highlight the differences between the two approaches. This could provide readers with a better understanding of the unique contributions of the proposed framework and how it compares to existing methods in the literature.

[1] Jiang, J., Zhou, K., Dong, Z., Ye, K., Zhao, W.X. and Wen, J.R., 2023. Structgpt: A general framework for large language model to reason over structured data. arXiv preprint arXiv:2305.09645.

**Reasons To Accept:**

The problem addressed in this manuscript is intriguing, and the proposed framework is technically sound. The results obtained from the experiments are promising, indicating the potential effectiveness of the proposed approach.

**Reasons To Reject:**

The paper would benefit from a more detailed explanation of the results, including how each step contributes to the overall performance and why the methods perform differently. This could help readers better understand the proposed framework and its effectiveness in addressing the problem of knowledge graph reasoning.

**Reproducibility:**

2: Would be hard pressed to reproduce the results. The contribution depends on data that are simply not available outside the author's institution or consortium; not enough details are provided.

**Reviewer Confidence:**

4: Quite sure. I tried to check the important points carefully. It's unlikely, though conceivable, that I missed something that should affect my ratings.

---

> ### Author Rebuttal · Authors · 2023-08-29
>
> Thank you for the time and effort spent in carefully reviewing our work. Please kindly find the responses below.
>
> **Reasons To Reject: The paper would benefit from a more detailed explanation of the results, including how each step contributes to the overall performance and why the methods perform differently. This could help readers better understand the proposed framework and its effectiveness in addressing the problem of knowledge graph reasoning.**
>
> In FactKG and MetaQA, there is no corresponding ground truth graph from the seed entity to the answer, making it challenging to quantitatively analyze each steps. Instead, we conducted an error analysis by extracting 100 incorrect samples each from FactKG, MetaQA-1hop, MetaQA-2hop, and MetaQA-3hop. The number of samples with errors at each step is shown in the table below. Overall, the number of errors that occurred during graph retrieval was the least among the three steps. This implies that when sentences are accurately divided, it's relatively easier to identify the relations within each sentence. Moreover, a comparative analysis of MetaQA-1hop, MetaQA-2hop, and MetaQA-3hop suggests that as the number of hops increases, the questions become more diverse. This diversity subsequently leads to increased errors in sentence segmentation.  We will add this analysis to the final version.
> | | FactKG | MetaQA-1hop | MetaQA-2hop | MetaQA-3hop |
> |---------|:---------:|:---------:|:---------:|:---------:|
> | Sentence Segmentation| 39| 3| 63| 100|
> | Graph Retrieval | 17| 4| 3| 0|
> | Inference | 44| 93| 34| 0|
>
>
>
> **Questions For The Authors: It is worth noting that StructGPT [1] appears to be very similar to KG-GPT, as both studies focus on the KGQA problem and use the same MetaQA data. Therefore, it would be interesting to highlight the differences between the two approaches. This could provide readers with a better understanding of the unique contributions of the proposed framework and how it compares to existing methods in the literature.**
>
> Our research shares similarities with StructGPT in terms of reasoning from structured data. However, while StructGPT identifies paths from a seed entity to the final answer entity within Knowledge Graphs, our approach involves retrieving the Graph and then obtaining the answer through inference in the end. One advantage of our framework over StructGPT is that finding an entity isn't our final output. This means our framework can be used not only for KGQA but also for tasks like KG-based fact verification. Moreover, we focus on a KG-based framework and KG-GPT outperforms StructGPT in MetaQA as described in the table below. Notably, we observe over a 13% performance difference in MetaQA-3hop. We will include a comparative analysis with StructGPT in the final version.
> | | MetaQA-1hop | MetaQA-2hop | MetaQA-3hop |
> |---------|:---------:|:---------:|:---------:|
> | StructGPT| 94.2| 93.9| 80.2|
> | KG-GPT | 96.3| 94.4| 94.0|

---

### Meta-Review · Area_Chair_wD5G · 2023-09-19

**Recommendation:** 3

**Metareview:**

This paper proposes KG-GPT,  a multi-purpose framework leveraging LLMs for tasks employing KGs. KG-GPT comprises three steps: Sentence Segmentation, Graph Retrieval, and Inference, each aimed at partitioning sentences, retrieving relevant graph components, and deriving logical conclusions. The sentence segmentation step partitions a sentence into discrete sub-sentences, each aligned with a single triple.  The graph retrieval step retrieves a potential pool of relations within the sub-sentences. In the final step, inference obtains graphs used to derive a logical conclusion, such as validating a given claim or answering a given question. Meanwhile, this paper evaluates KG-GPT using KG-based fact verification and KGQA benchmarks, which fills the gap in complex reasoning on knowledge graphs for LLMs.

---

### Decision · Program_Chairs · 2023-10-07

**Decision:**

Accept-Findings

**Comment:**

This paper proposes KG-GPT,  a multi-purpose framework leveraging LLMs for tasks employing KGs. KG-GPT comprises three steps: Sentence Segmentation, Graph Retrieval, and Inference, each aimed at partitioning sentences, retrieving relevant graph components, and deriving logical conclusions. The sentence segmentation step partitions a sentence into discrete sub-sentences, each aligned with a single triple.  The graph retrieval step retrieves a potential pool of relations within the sub-sentences. In the final step, inference obtains graphs used to derive a logical conclusion, such as validating a given claim or answering a given question. Meanwhile, this paper evaluates KG-GPT using KG-based fact verification and KGQA benchmarks, which fills the gap in complex reasoning on knowledge graphs for LLMs.